# Circulation of Pestiviruses in Small Ruminants from Greece: First Molecular Identification of Border Disease Virus

**DOI:** 10.3390/vaccines11050918

**Published:** 2023-04-28

**Authors:** Ilias G. Bouzalas, Athanasios I. Gelasakis, Taxiarchis Chassalevris, Evangelia D. Apostolidi, Fotis Pappas, Loukia Ekateriniadou, Evridiki Boukouvala, Antonios Zdragas

**Affiliations:** 1Veterinary Research Institute, Hellenic Agricultural Organization DIMITRA (ELGO-DIMITRA), Campus Thermi, 57001 Thessaloniki, Greece; 2Department of Animal Science, School of Animal Biosciences, Agricultural University of Athens, Iera Odos 75, 11855 Athens, Greece; gelasakis@aua.gr

**Keywords:** pestiviruses, small ruminants, border, sheep, goats, Greece, persistently infected

## Abstract

The incidence of small ruminant pestivirus infections in Greece remains unknown as they have not been diagnosed in the country since 1974 when the most recent Border Disease Virus (BDV) outbreak was reported. The objective of our study was to explore the possible occurrence of pestiviral infections among sheep and goat farms in Greece and to further determine the variants of major concern. Thus, serum samples were collected from 470 randomly selected animals belonging to 28 different flocks/herds. ELISA on p80 antibody revealed the existence of seropositive animals in four out of the 24 studied sheep flocks, whereas all the goats in the four studied herds were seronegative. Viral RNA and antigens were detected in two sheep out of the four seropositive flocks by RT-PCR and ELISA, respectively. Sequencing and phylogenetic analysis showed that the newly identified Greek variants were closely related to the strains of the BDV-4 genotype. One of the BDV-positive sheep demonstrated the diagnostic profile of a persistently infected (PI) animal, providing additional information regarding the source of the infection. This is the first molecular identification of BDV isolates in Greece. Our findings indicate that BDV infections are likely to remain undiagnosed, highlighting the need for further epidemiological studies and active surveillance programs to determine the prevalence and impact of BDV infections on a countrywide level.

## 1. Introduction

Belonging to the *Flaviviridae* family, the genus *Pestivirus* includes four main species, namely bovine viral diarrhea virus -1, -2 (BVDV-1, BVDV-2), classical swine fever virus (CSFV) and border disease virus (BDV) [1,2]. Following the motif of other pestiviruses that are not considered to be highly host-specific, BVDV mainly affects cattle but has also been identified in goats, while BDV has been detected in a wide variety of animal species, with sheep being the dominant host. Both ruminant pestiviruses, BVDV and BDV, are distributed globally, causing substantial monetary losses in the livestock sector due to their impact on the health status and reproductive efficiency of the infected animals [3,4].

BDV is an enveloped virus, and its genome consists of a single-stranded, positive polarity RNA composed of approximately 12,500 nucleotides. This genome contains a single open reading frame (ORF) that encodes four structural proteins, the capsid (C) and three envelope glycoproteins (Erns, E1, and E2), and seven to eight non-structural proteins (Npro, p7, NS2–3, NS4A, NS4B, NS5A, and NS5B), which are flanked by 5′ and 3′ large untranslated regions (UTRs) [5].

In sheep, clinical manifestation of Border Disease (BD) may include barren ewes, abortions, and stillbirths, as well as the birth of weak lambs with hairy fleece, abnormal body conformation, and tremors [5]. In goats, field cases of clinical disease caused by pestiviruses are rarely reported and have been mostly characterized by abortion and poor viability of newborn kids [5]. However, recovery of pestiviruses from naturally infected goats is rare [6,7,8,9].

Infected animals can be detected by various laboratory techniques. ELISA protocols detecting specific antibodies against ruminant pestiviruses are broadly used, either to confirm the infection or to estimate the seroprevalence in a population [10,11]. However, ELISA is inadequate for characterizing the type of responsible pestivirus, and thus, Virus Neutralization Tests (VNTs) or molecular techniques have to be applied for further discrimination [12]. VNTs are laborious and require advanced virological facilities, whereas reverse transcriptase PCR is commonly used due to its simplicity and accurate antigenic determination of the viral strain [13]. Moreover, PCR testing may reveal the presence of transiently or persistently infected (PI) animals, with the latter being the most potent source of infection.

Persistent infections in newborns are attributed to the fact that fetuses whose immune system is immature get exposed to the virus [5]. Specifically, pregnant animals usually are asymptomatic or exhibit only mild clinical signs and are immune to reinfection since they develop neutralizing antibodies. On the contrary, fetuses infected before the onset of immune competence are prone to fetal death due to uncontrollable viral replication. However, the lambs that survive after being infected in early gestation are characterized by multiorgan viral spread. These animals appear to become immunotolerant to the infecting virus strain, usually develop normally and remain persistently infected for life. Persistently infected animals may not exhibit any clinical signs, while often, they are characterized by a shortened lifespan and stunted growth [14]. Moreover, BDV persistently infected lambs may show the pathognomonic symptoms of rhythmic tremor, ataxia, and an abnormal, hairy fleece (hairy shaker syndrome) (longer and finer coat) [5,15,16]. Persistently infected animals excrete large viral loads constantly and are responsible for the continuous circulation of the virus in the host population [14]. Therefore, the identification of persistently infected animals is of major significance when designing appropriate prevention strategies against the disease. In addition, accurate diagnosis of persistently infected animals is important since there are no proven effective vaccines against BDV. As has been summarized in a recent publication, killed whole-virus vaccines have been developed in some countries but have never been commercialized, whereas the occasional attempts to use BVDV vaccines have been proven to be partially effective because even though the two viruses are antigenically related, they do not seem to elicit full cross-protective immune responses [17].

A multitude of epidemiological studies has shown that BDV infections have a global distribution [17]. According to their genetic heterogeneity, the different BDV strains collected around the world have been classified into at least eight genotypes, from BDV-1 to BDV-8. Additionally, BD-like syndromes can also be caused by other ovine pestiviruses (Aydin-like, Tunisian and Tunisian-like viruses) and a novel ovine pestivirus (OVPV) related to classical swine fever virus (CSFV) [17].

In Greece, a small ruminant pestivirus outbreak (BDV) was reported for the first time in 1974 [18]. The report was based on compatible clinical findings and the limited serological tests that could be applied at that time. Since then, there have been no further studies investigating the occurrence of pestiviral infections in Greece, despite the high number of small ruminants kept in the country. The objectives of our study were to investigate the potential occurrence of pestiviral infections in sheep and goat flocks/herds in Greece and to further assess the possible sources of infection.

## 2. Materials and Methods

### 2.1. Sample Collection Strategy

Between 2018 and 2020, 24 sheep flocks and four goat herds from various geographical regions of Greece (Table 1, Figure 1) were enrolled in the study. Initially, from these flocks/herds, 470 blood samples were collected from randomly selected, healthy, adult (over 12 months of age) sheep and goats. Collected samples were provided by veterinarians as part of standard veterinary practice. The sample size was estimated based on the number of adult animals in each flock/herd; namely, in flocks/herds with less than 200 adult animals, the sample size was estimated at 10%, whereas from each flock/herd with more than 200 animals, 20 samples were collected. Blood samples were collected in Vacutainer-type tubes without anticoagulant, and following centrifugation at 3000× *g* for 10 min, the separated serum was transferred to sterile 1.5 mL capacity tubes and stored at −20 °C until it was further analyzed.

In order to discriminate the transiently infected from the persistently infected animals, additional serum samples were collected from the infected animals two months later.

### 2.2. Serological Detection of Specific Ruminant Pestivirus Antibodies by ELISA

The serological detection of pestiviruses in the studied population was conducted using a commercially available ELISA kit [PrioCHECK^TM^ Ruminant BVD and BD p80 Ab Serum Kit (Applied Biosciences)], according to the manufacturer’s instructions for pooled serum samples. Briefly, 28 pooled serum samples were created by mixing 40 μL serum per animal in each pooled sample. Each flock/herd was represented by one pooled sample containing the sera of up to 20 animals. Interpretation of the results suggested that a competition percentage of each pooled sample superior or equal to 65% was indicative of a high probability that PI animals were present within the flock/herd. On the contrary, if the competition percentage was below 65%, there was a low possibility of viral circulation within the flock/herd.

Consequently, individual serum samples included in the seropositive pooled samples were further analyzed using the same ELISA kit but following the manufacturer’s instructions for the individual serum samples. In that case, individual samples were characterized as strongly positive, weakly positive, and negative when the competition percentage was superior or equal to 80%, between 50 and 79%, and below 49%, respectively.

### 2.3. Real-Time RT-PCR and Antigen (Ag) ELISA

To identify potential asymptotic carriers of pestiviruses, all seronegative individual samples originating from the seropositive flocks/herds were tested using a commercially available real time RT-PCR kit [LSI VetMAX^TM^ BVDV 4ALL kit (Thermofisher Scientific, Waltham, MA, USA)]. According to the manufacturer’s instructions, an internal positive control (IPC) was included during the nucleic acid purification process in order to ensure the efficient RNA isolation. Viral RNA from the tested sera was purified using the PureLink^TM^ Viral RNA/DNA Mini Kit (Thermofisher Scientific, Waltham, MA, USA).

Serum samples from animals that reacted positive (n = 2) in the real-time RT-PCR protocol described above were additionally tested by a commercial capture ELISA, the IDEXX BVDV Ag/Serum Plus Test (IDEXX, Berne, BE, Switzerland), following the manufacturer’s instructions. The latter—often termed Ag ELISA—was used to validate the extra-label performance of the kit for detecting small ruminant pestiviruses since the official recommendation is directed towards identifying the BVD virus in cattle. Additional real-time RT-PCR negative samples (n = 92) were also included in every run of the Ag ELISA.

For identifying the putative PI animals, additional serum samples were collected in a two-month interval period from the animals that initially reacted positively when tested with the aforementioned methods (real-time RT-PCR and Ag ELISA). These additional serum samples were also tested by real-time RT-PCR and Ag ELISA.

### 2.4. Phylogenetic Analysis

Phylogenetic analysis was based on the genetic determination of the 5′ untranslated region (5′UTR) of the viral genome due to its ability to detect and characterize a very wide spectrum of pestivirus isolates [19]. A cDNA synthesis was carried out from the real-time RT-PCR positive RNA samples using Maxima H Minus First Strand cDNA Synthesis Kit (Thermofisher Scientific, Waltham, MA, USA) as described by the manufacturer’s protocol. Amplification of cDNAs by PCR was performed using the primer pairs 324/326 for the *Pestiviruses* 5′UTR genomic region [17]. Ultrapure water was used in every run, instead of cDNA, as a negative control. Fifty microlitres of each PCR product were analyzed on 1% agarose (Sigma-Aldrich, St Luis, MO, USA) gel containing ethidium bromide (Sigma-Aldrich, St Luis, MO, USA). The fragments were visualized using a UV transilluminator. Amplicons of the expected sizes (288 bp) were excised from the agarose gel and recovered using the PureLink^TM^ Quick PCR Purification Kit (Thermofisher Scientific, USA) and stored at −20 °C.

Amplified products were sequenced in both directions with primers 324 and 326 using an ABI 3500 Genetic Analyzer (Thermofisher Scientific, Waltham, MA, USA) and the BigDye terminator sequencing protocol. Nucleotide sequences of representative pestivirus members were aligned to the sequences obtained in our study using the MEGA 7 software [20]. Phylogenetic distances were calculated on a neighbor-joining tree using the Jukes–Cantor model with 1000 bootstrap replicates.

## 3. Results

### 3.1. Antibody Prevalence

Antibodies against ruminant pestiviruses were detected in 40 out of 470 serum samples examined by ELISA (8.51%). None of the studied goats (n = 53) was found seropositive; hence, all the 40 seropositive samples originated from the studied sheep population (n = 417), resulting in a seroprevalence equal to 9.59% in this species. The latter corresponds to a seroprevalence of 16.66% (4/24) at the flock level (all 40 seropositive sheep originated from four different herds). Of those four positive sheep flocks, the within-flock seroprevalence was 70% (7/10), 60% (12/20), 85% (17/20), and 40% (4/10) in Flocks No 1, 12, 13 and 28, respectively (Table 1). Regarding the Optical Density values (ODs) detected by ELISA in the pooled sera samples (one pool per flock), it was noticed that Flocks No 1, 12, and 13 showed a competition percentage above 65%, indicating the circulation of the virus within them.

The majority of the seropositive flocks were located in Northern Greece. In particular, two seropositive flocks (No 1 and 28) were located in the prefecture of Kozani, and one (No 12) was located at Serres, in a territory quite distant from Kozani. Finally, the fourth positive flock (No 13) was located in central Greece, at the prefecture of Fthiotida (Figure 1).

### 3.2. Viral Detection with Real Time RT-PCR and Ag ELISA

In order to identify putative BDV-infected sheep within the seropositive flocks, all serum samples from seronegative sheep (n = 20) were tested with real-time RT-PCR targeting the viral genome. Our study revealed the presence of viral RNA in two out of the 20 seronegative samples (10%). One positive sample, encoded 136LamiGR/19, was derived from flock No 13 and showed a Ct value of 28.79 (Figure 2), while the other one with a Ct value of 30.48 belonged to herd No 28 and was encoded as 10KozaGR/20. These two samples also reacted positively when tested with Antigen (Ag) ELISA. Two months later, only one of the two corresponding sheep was available, and the obtained serum sample was found positive when both techniques (real-time RT-PCR and Ag ELISA) were applied for the second time, confirming the characterization of the specific sheep as PI animals.

### 3.3. Phylogenetic Analysis

Nucleotide sequences derived from the two RT-PCR positive animals were used for BLAST searching against GenBank to identify similarities with already submitted sequences. The comparison between consensus sequences and representative members of other pestiviruses was visualized on a neighbor-joining tree using MEGA 7 software (Figure 3). Both sequences were clustered in the group of BDV 4. According to the data derived from BLAST, the 136LamiGR/19 (GenBank accession number OQ077189) was closer to the strain Chamois-Spain02 isolated from Chamois in the Pyrenees mountains (AY641529) and the BDV 5 French isolate Aveyron (KF918753) revealing an identity score of 91.6% and 90.9%, respectively, with a coverage of 98.0%. The second Greek isolate named 10KozaGR/20 (GenBank accession number OQ077190) revealed an identity score of 96.1% with the Spanish strain 0502234 (EU711348) when the coverage was 95% and an identity score of 99.2% with the Spanish strain M3 (DQ275626) when the coverage was 81%. A pairwise comparison of the 5′UTR nucleotide sequences between the two Greek strains identified in our study (136LamiGR/19 and 10KozaGR/20) revealed a percentage similarity of ~89%.

## 4. Discussion

The first report of small ruminant pestivirus infection in Greece was in 1974 when an extensive BDV epizootic affected more than 1000 sheep flocks in North-Eastern Greece (East Macedonia and Thrace) [18]. Since then, and even though there have been cases with BD-compatible clinical signs, laboratory confirmation has never been achieved. Therefore, no data regarding the occurrence of pestivirus infections in Greece is available. This is the first study after almost 50 years that confirms the circulation of small ruminant pestiviruses strains in Greek small ruminant farms.

Our findings indicate that the apparent prevalence of seropositive sheep and sheep herds in continental Greece was 9.59% and 16.66%, respectively. The number of the assayed samples and the studied flocks was not large enough for a definitive epidemiological estimation of the true prevalence; however, the results of our study reveal that pestiviruses should be taken into consideration when determining the differential diagnosis of reproductive problems in Greek sheep. Although it is likely that pestivirus infections are not endemic in most of the regions in Greece, our findings indicate that further countrywide epidemiological studies are necessary to determine their spatial spreading. However, the identification of the viral antigens in animals, even though in a very limited number (n = 2), cannot be ignored as they provide evidence for a potential more extensive transmission in the future, within and between farms. Control strategies, including active and passive surveillance systems, monitoring of ruminants at risk on an evidential basis, and laboratory confirmation of the BD-free status of imported breeding stocks, should be implemented in order to reduce the risk of spreading the disease. Similar measures have been proposed in other countries, where the disease has been prevalent for years [10,21], and they are likely to be effective in Greece, taking into consideration that the spreading of the disease seems to be still limited.

Interestingly, molecular testing revealed the presence of one BDV PI animal within the investigated herds providing complementary diagnostic information regarding the viral transmission dynamics. Phylogenetic analysis of selected samples indicated that the flocks under study were infected by strains that were closely related to BDV-4 genotype. BDV-4 genotype has been mainly reported in France and Spain (chamois and sheep respectively) [22,23,24], supporting our previous, based on serology, assumption, that BDV infections are not endemic in Greece. The latter seems to be consistent with the breed origin of the infected animals. As indicated in Table 1, all infected flocks in this study consisted of animals of the French breed Lacaune, while flocks with animals of indigenous breeds, such as Chios, were found negative in all cases. Specifically, the infected flock in which the PI animal was found had imported Lacaune hoggets one year before our diagnosis. Unfortunately, the ear tag of the PI animal was missing, and therefore it was not possible to verify whether it was among the imported ones or it was born on the farm. In any case, this is the first report of molecular detection and characterization of BDV variants in Greece and highlights the necessity of taking rapid countermeasures against the disease.

Several different protocols have been described for the phylogenetic comparison of pestiviruses. Most of them have been based on the comparison of the regions encoding the non-structural proteins Npro, the structural protein E2 and the 5′ untranslated region (5′UTR) [8,19,25,26]. The non-structural protein, Npro, codes for the N-terminal auto protease, which has no counterpart in other flaviviruses, whereas the E2 protein plays a major role in virus attachment and entry and is also important in inducing neutralizing antibody production [8,26]. The 5′UTR seems to be the most conserved region of the viral genome, and its genetic determination using the specific primer pair 324/326 provides the ability to characterize a high variety of ruminant pestiviruses [19]. Therefore, the last method was selected for the phylogenetic comparison of our strains. However, it has been well documented that the classification of pestiviruses might become complicated in some cases [27]. Thus, additional studies regarding the genetic characterization of other than 5′UTR regions are required to ensure the proper classification of the Greek variants.

A shortcoming of the present study could be the fact that VNT was not used to evaluate the circulation of other small ruminant pestiviruses. Such a technique could be applied in all seropositive animals with a special focus on the ones originating from flocks where viral antigen could not be detected. In order to expand our knowledge on ruminant pestiviruses in Greece, the next planned step is to test with VNT assays the sera collected from seropositive animals of this study and additional samples that have been collected in the meantime. However, the phylogenetic analysis revealed that antigen-carrier animals had an active BDV-4 infection, indicating that currently, BDV is likely to be the main pestivirus of concern in the Greek sheep population.

A remarkable finding of our study was that the commercial capture ELISA [IDEXX BVDV Ag/Serum Plus Test (IDEXX, Switzerland)] was able to detect all positive BDV sheep, even though it is recommended for the diagnosis of BVD virus in cattle. Indeed, the aforementioned ELISA kit had 100% concordance with real-time RT-PCR results, indicating the potential to be used as an extra-label diagnostic tool for BDV infections after further validation of its diagnostic performance; for efficient validation, it is necessary to test more samples from antigen-positive animals collected from a higher number of farms across the country and under various circumstances. This finding is contradictory to other studies claiming that ELISA is less sensitive than real-time RT-PCR for detecting BDV-4-infected animals [25]. Although our data have been based on a very limited number of animals, such differences might be attributed to the different peptides used in each ELISA protocol along with the differences in the performance of the applied molecular protocols.

In conclusion, the results of the present study indicate that the circulation of small ruminant pestiviruses in Greek sheep is currently low but high enough to be considered an emerging issue of concern. The detection of specific anti-pestivirus antibodies is currently the most common and feasible approach to investigate the spreading of pestivirus infections since there are no available registered vaccines against the disease. However, PCR can be considered a useful complementary diagnostic tool for the confirmation of the diagnosis and the discrimination of PI animals, as it is a low-cost and minimally laborious technique. The combination of the above techniques could set the base for disease monitoring strategies that should be conducted in the entire ruminant population of Greece to eliminate the risk of the virus spreading. In addition, the infectivity status of imported animals should be taken under special consideration.

## Figures and Tables

**Figure 1 vaccines-11-00918-f001:**
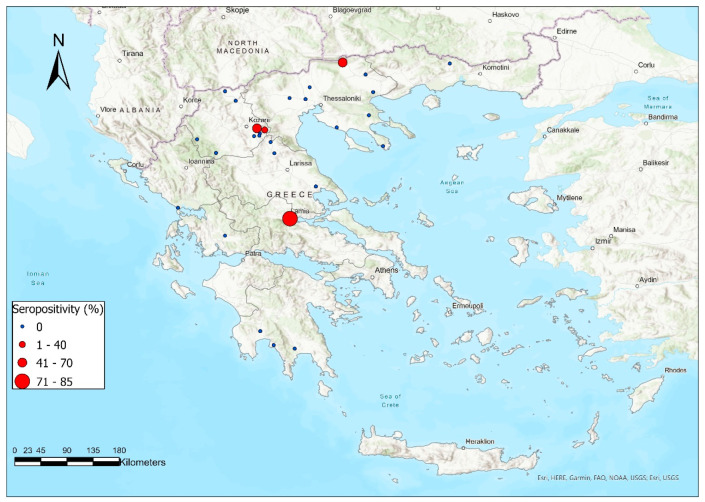
Geographical distribution and serological profile of the flocks/herds included in our study.

**Figure 2 vaccines-11-00918-f002:**
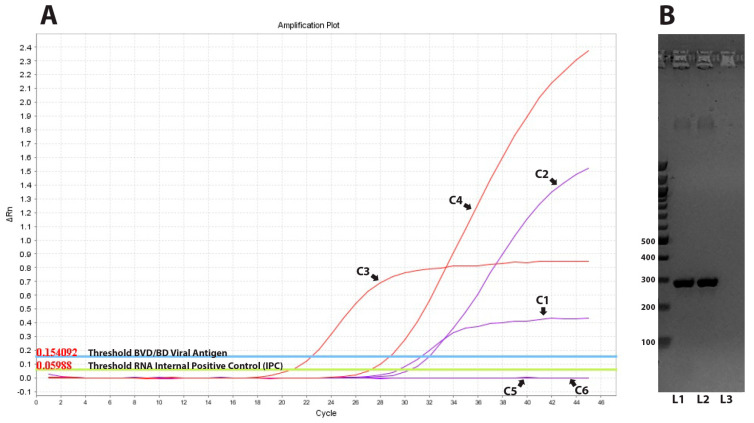
Molecular detection of ruminants’ pestiviruses: (**A**) Real time RT-PCR results. C1 = amplification curve of internal positive control (IPC) from the positive control of the kit; C2 = amplification curve of BVD/BD viral RNA from the positive control of the kit; C3 = amplification curve of internal positive control (IPC) from the positive sample of herd No 13; C4 = amplification curve of BVD/BD viral RNA from the positive sample of herd No 13; C5 and C6 = amplification curves of negative control samples. (**B**) Electrophoresis of RT-PCR products amplified by the 324/326 primers pairs. L1 = positive sample of herd No 13; L2 = positive sample of herd No 28; L3 = negative control. Numbers on the left indicate the molecular weight in base pairs (bp).

**Figure 3 vaccines-11-00918-f003:**
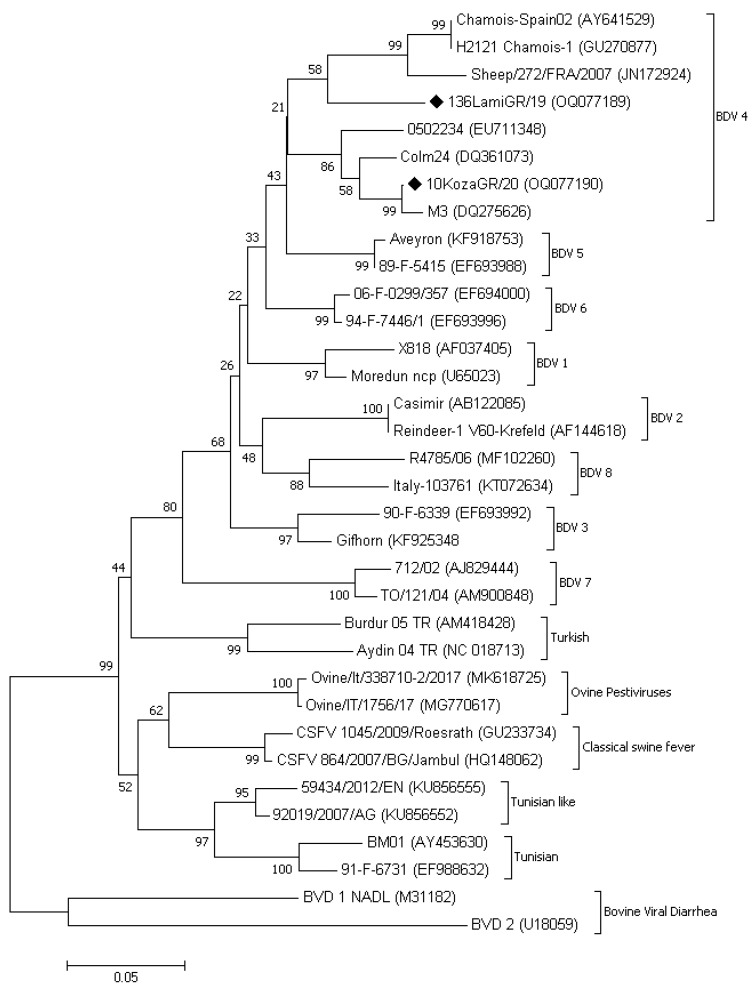
Phylogenetic comparison based on the 5′ UTR viral genome locus between the variants of our study and representative members of other pestiviruses, with emphasis on Border Disease Virus (BDV) strains. The comparison was based on a neighbor-joining tree using MEGA 7 software. GenBank accession numbers are given in brackets, and p-distances are indicated by scales at the bottom. The Greek variants detected in our study were marked by a rhombus symbol.

**Table 1 vaccines-11-00918-t001:** Frequencies of BDV positive flocks/herds and animals.

Flock/ Herd ID	Area	Species	Breed *	ELISA Positive/Tested Animals (Seropositivity%)	PCR Positive/Tested Animals (Genotype)
1	Kozani	Sheep	Lacaune	7/10 (70%)	0/3
2	Kozani	Sheep	Lacaune	0/10 (0%)	0/0
3	Kozani	Sheep	Chios cross	0/10 (0%)	0/0
4	Xanthi	Sheep	Assaf cross	0/20 (0%)	0/0
5	Lakonia	Sheep	Lacaune cross	0/20 (0%)	0/0
6	Serres	Sheep	Assaf cross	0/19 (0%)	0/0
7	Messinia	Sheep	Chios cross	0/20 (0%)	0/0
8	Thessaloniki	Sheep	Chios	0/20 (0%)	0/0
9	Kozani	Goat	Alpine cross	0/20 (0%)	0/0
10	Florina	Goat	Saanen cross	0/7 (0%)	0/0
11	Magnisia	Sheep	Chios	0/20 (0%)	0/0
12	Serres	Sheep	Lacaune	12/20 (60%)	0/8
13	Fthiotida	Sheep	Lacaune	17/20 (85%)	1/3 (BDV-4) **
14	Florina	Goat	Alpine cross	0/16 (0%)	0/0
15	Aitolokarnania	Sheep	Chios	0/20 (0%)	0/0
16	Chalkidiki	Sheep	Chios	0/20 (0%)	0/0
17	Kilkis	Sheep	Chios	0/20 (0%)	0/0
18	Preveza	Sheep	Chios	0/20 (0%)	0/0
19	Pella	Sheep	Chios	0/20 (0%)	0/0
20	Grevena	Sheep	Lacaune cross	0/20 (0%)	0/0
21	Grevena	Sheep	Assaf cross	0/10 (0%)	0/0
22	Chalkidiki	Goat	Alpine cross	0/10 (0%)	0/0
23	Messinia	Sheep	Lacaune cross	0/20 (0%)	0/0
24	Larissa	Sheep	Chios cross	0/20 (0%)	0/0
25	Larissa	Sheep	Lacaune cross	0/20 (0%)	0/0
26	Chalkidiki	Sheep	Assaf	0/19 (0%)	0/0
27	Chalkidiki	Sheep	Lacaune cross	0/9 (0%)	0/0
28	Kozani	Sheep	Lacaune cross	4/10 (40%)	1/6 (BDV-4) ***
TOTAL				40/470 (8.51%)	2/20 (BDV-4)

* Breed was determined based on the dominant phenotype of the flock/herd, cross = crossbred; ** Confirmed as a PI animal; *** na = not available for re-checking.

## Data Availability

Data is contained within the article.

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
