# Peer review of "Circulation of Pestiviruses in Small Ruminants from Greece: First Molecular Identification of Border Disease Virus"

_vaccines, 2023, doi:10.3390/vaccines11050918_

Round 1

Reviewer 1 Report

The article submitted by Ilias et al on vaccines investigates the seroprevalence and molecular identification of BDV virus infections in ruminants in Greece. I would take the opportunity to appreciate the authors for taking an effort to identify BDV in the country and this should be the first level for active surveillance of the same in future. The techniques detailed in the article is good enough for a surveillance program.  The article is written very well and methodology followed for sample collection is reasonable.  I have very few minor comments.

Abstract

(1) Rephrase the sentence “Antibody ELISA revealed the existence of seropositive animals in 4 out of the 24 studied sheep flocks, whereas all the goats in the 4 studied herds were seronegative” to “ELISA on p80 antibody revealed the …..”

(2) Rewrite the sentence to “Viral RNA and antigens were detected in 2 sheep out of the 4 seropositive flocks by RT-PCR and ELISA respectively.

Materials and methods

(3) It would be good to mention why you chose 5’UTR region for RT-PCR analysis.

Reviewer 2 Report

Major comments

1) Title - it sounds inaccurate ("isolates" refer to laboratory conditions) and should be changed.

2) More details should be provided about sample collection strategy.

3) lines 114-122 - these sentences are somewhat unclear and should be improved.

4) Line 164 - "antigen-positive"? Or virus-positive. "Both techniques" include PCR? Thus, not antigen but genome? Please, correct this sentence.

5) Lines 263-264 - the number of investigated animals (n=2) appears unsuitable to support this hypothesis.

6) Discussion - data about BDV in neighboring countries (if any) should be provided.

7) Conclusion - Authors should report data about the putative origin of BDV (France?) at this point.
